# Differential insulin response characteristics of graphene oxide–gold nanoparticle composites under varied synthesis conditions

**Qian Zhang, Yanjun Pan, Jin Pan, Zhichen Wang[ID], Ruyi Lu, Jing Sun, Jingyi Feng[ID]***

Key Laboratory of Clinical Evaluation Technology for Medical Device of Zhejiang Province, Department of Clinical Engineering and Material Supplies, The First Affiliated Hospital, Zhejiang University School of Medicine, Hangzhou, P.R. China

* feng.jingyi@zju.edu.cn

**Data Availability Statement:** All relevant data are within the manuscript and its Supporting Information files.

## Abstract

The structural alterations in the constituent materials of nanocomposites such as graphene nanocomposites typically induce changes in their properties including mechanical, electrical, and optical properties. Therefore, by altering the preparation conditions of nanocomposites and investigating their responsiveness to basic biomolecules (such as proteins), it is possible to explore the application potentials of the composites and guide development of new nanocomposite preparation. In this study, different composites of graphene oxide and gold nanoparticles (AuNPs/GO) were obtained by varying the volumes of reducing agents used in the one-pot hydrothermal method. Insulin was chosen as a basic protein to study the response characteristics of AuNPs/GO under different preparation conditions. Optical responses of these composites to pure insulin and various commercial insulin types were all explored for the first time. The results indicated that AuNPs/GO could optically respond to insulin, including pure insulin and various types of commercial insulin, and changes in the preparation conditions could really influence this response. Moreover, optimal preparation conditions could be determined by an optical method for the largest responses of the nanocomposites to insulin. Based on previous research and the results of this study, it is speculated that the responses of AuNPs/GO to insulin may attribute to glutamic acids, asparagines, and glutamines on insulin, which may interact with AuNPs/GO, particularly with the AuNPs in the composites. Besides, the AuNPs/GO could exhibit relatively stable responses to various commercial insulin types and detect the concentration of specific branded commercial insulin with smaller errors. In summary, this study demonstrated the application potential of AuNPs/GO in areas such as drug testing and production, while also furnishing an experimental foundation and direction for further applications of AuNPs/GO in biosensing and biomolecule detection.

## Introduction

As a member of the graphene family, graphene oxide (GO) is commonly used in the fabrication of nanocomposites, owing to its ease of preparation and processing, high hydrophilicity,

**Funding:** This research was funded by Zhejiang Provincial Natural Science Foundation of China, Grant No. LQ20C100001, and Zhejiang Provincial Department of Education General Research Project of China, Grant No. Y201941826. The funders had no role in study design, data collection and analysis, decision to publish, or preparation of the manuscript.

**Competing interests:** The authors have declared that no competing interests exist.

two-dimensional structure with various functional groups, tunable optical properties, and biocompatibility [1–3]. Among the currently prepared graphene oxide-based composites, those garnering the most extensive attention involve the integration of graphene oxide with gold nanoparticles (AuNPs) to form composites (AuNPs/GO) [4–6]. A search on Web of Science database using 'graphene oxide' and 'gold nanoparticles' as topics revealed a total of 9,253 relevant articles and review articles published between 2014 and 2023. The high specific surface area, tunable surface chemical properties, localized surface plasmon resonance (LSPR), high sensitivity, and other characteristics of AuNPs make them promising in the field of sensing, particularly in label-free sensing [7–9]. The combination of AuNPs and graphene oxide provides a high density of active binding sites for analyte molecules, resulting in higher sensitivity [10]. Zhu et al. fabricated a periodically tapered structure-based gold nanoparticles and graphene oxide—immobilized optical fiber sensor to detect ascorbic acid, and found that use of AuNPs and GO could enhance the sensing performance of the sensor [11]. Mohsin et al. developed an innovative sensing platform employing gold nanoparticles and reduced graphene oxide (rGO) for ultrasensitive Hepatitis B virus detection. Their investigation revealed that the strategic incorporation of AuNPs with rGO provides multifaceted advantages, including preventing nanosheet structural aggregation, substantially increasing the effective surface area, significantly lowering detection limits, and markedly improving biocompatibility [12]. Yang et al. conducted a sophisticated localized surface plasmon resonance (LSPR)-based tapered fiber sensor utilizing gold nanoparticles and graphene oxide. Their research elucidated that LSPR intensity and wavelength are critically modulated by the morphological and structural characteristics of gold nanoparticles, specifically the nanoparticle size, morphological configuration, and surrounding medium's refractive index [13, 14]. Consequently, the conjugation of AuNPs with GO induces substantial modifications in sensitivity and LSPR properties, presenting promising opportunities for advanced biochemical sensing and diagnostic applications.

Extensive investigation has been conducted regarding various preparation methodologies and applications in the biosensing domain for AuNPs/GO composites [15, 16]. Notably, most studies have emphasized diverse preparation techniques and more sensitive detection strategies for these composites, typically focusing on fixed nanoparticle sizes and specific biomarkers [17–19]. However, a comparative analysis of the response characteristics exhibited by various nanocomposites derived from different preparation protocols to the same biological molecule has been relatively rare. Significant structural changes in each constituent material can considerably influence the properties of nanocomposites, thus impacting their potential applications [20–22]. For example, Xue-Peng Wei et al. prepared multi-walled carbon nanotubes@titanium dioxides with carboxymethyl chitosan nanocomposite (MWCNTs@TiO$_2$/CMCS), and found that changes in the amount of MWCNTs or TiO$_2$ could alter the redox property of the composite [23]. Therefore, research on the biological detection properties of AuNPs/GO composites prepared under different protocols holds significant importance and offers valuable insights for subsequent studies.

Meanwhile, investigations into the fundamental biological response characteristics of nanocomposites help to further elucidate their application potentials and guide future research endeavors including performance and preparation method optimization [24, 25]. As a basic protein, insulin is important and highly studied [26]. Existing insulin detection methods can be classified into four major categories: immunoassay, chromatography, electrochemical methods, and optical methods [27]. The integration of nanomaterials with these methods has become a hot area in recent years. Incorporating nanomaterials provides additional functional active sites and enables highly sensitive, personalized sensing strategies [28]. Commonly reported nanomaterials used for insulin detection include carbon nanotubes, gold/silver

nanoparticles, silicon-based materials, and others [27, 29, 30]. For instance, Bisker et al. utilized corona phase molecular recognition to adsorb PEGylated lipids onto single-walled carbon nanotubes and detected insulin by monitoring changes in fluorescence emission [31]. Khan et al. fabricated a composite of silver/$ZnIn_2S_4$/reduced graphene oxide and gold-modified silica nanoparticles, which served as the electrochemiluminescent donor and acceptor, respectively, enabling the sensitive and accurate detection of insulin [32]. However, most detection methods employ insulin-sensitive materials to enhance sensitivity and specificity, with the preparation methods for the utilized nanomaterials being fixed. There is a paucity of research on the insulin response characteristics of nanomaterials, especially those prepared using different methods.

Our previous research has already examined the responsive characteristics of the AuNPs/GO nanocomposite to bioactive small molecules, specifically amino acids [33]. The results demonstrated that, under fixed preparation conditions, this nanocomposite exhibited visible light responses only to acidic amino acids and cysteine among common amino acids. This study further explored the responsive characteristics of AuNPs/GO to insulin under varied preparation conditions for the first time. Chosen pure insulin and different types of commercial insulin as target analytes, various AuNPs/GO were prepared by altering the amount of reducing agent during the synthesis process and their optical responses to target analytes were investigated. The primary objective of this research was to establish a foundation for further improvement and enhancement of AuNPs/GO composites and their applications in biodetection.

## Materials and methods

### Materials and instruments

Graphene oxide dispersion (concentration: 1 mg/mL, thickness: 1–5 layers, solvent: water, CAS number: 7440-44-0) was purchased from Jiangsu XFNANO Materials Tech Co., Ltd. Tetrachloroauric acid trihydrate ($HAuCl_4 \cdot 3H_2O$, 1g, $\geq$99.9% trace metals basis, CAS number: 16961-25-4) and insulin (derived from bovine pancreas, 100 mg, CAS number: 11070-73-8) were obtained from Shanghai Aladdin Bio-Chem Technology Co., Ltd. Sodium citrate powder was purchased from China National Pharmaceutical Group Corporation. NovoRapid® (Insulin Aspart Injection), Novolin® R (Human Insulin Injection), Novolin® N (Isophane Protamine Human Insulin Injection), Levemir® (Insulin Detemir Injection), and Novolin® 30R (Mixed Protamine Human Insulin Injection (30R)) were all purchased from Novo Nordisk Pharma (China) Co., Ltd.

Magnetic stirring was performed using an IKA magnetic stirrer (C-MAG HS 7). Transmission electron microscopy (TEM) images were acquired using a JEM-1400flash transmission electron microscope from JEOL Ltd. Ultraviolet and visible (UV–Vis) absorption spectroscopy was conducted using an Ocean Optics UV-Vis spectrometer (USB2000+). Electrochemical impedance spectroscopy (EIS) measurements were conducted using a CHI660E electrochemical workstation from Shanghai Chenhua Instruments Co., Ltd., employing a three-electrode system comprising a gold working electrode, an Ag/AgCl reference electrode, and a platinum wire counter electrode. Solution agitation was achieved using an IKA Lab-dancer with a stirring speed of 2800 r.p.m.

### Preparation and characterization of AuNPs/GO

The preparation of the composites followed the procedure outlined in a previous study [33]. In brief, 3 mL of graphene oxide, 1.225 mL of 1% (w/w) chloroauric acid, and 150 mL of deionized water were mixed and stirred at 80˚C. Then, 1–2.5 mL of 1% (w/w) sodium citrate

solution was rapidly added. The mixture was further stirred and heated until the reaction was completed, resulting in the formation of various composites comprising AuNPs and GO.

For the characterization of AuNPs/GO composites, this study employed several techniques, including transmission electron microscopy (TEM), UV–Vis spectroscopy, and EIS. Specifically, for TEM analysis, a uniform 20 µL drop of the composite solution containing the AuNPs/GO composite (with a reducing agent volume of 1.8 mL) was carefully deposited onto a copper grid and air-dried before imaging.

In UV–Vis spectroscopy and electrochemical impedance spectroscopy, the study compared AuNPs/GO solution prepared with 1.8 mL of reductant with a pure AuNP solution (prepared using the same steps as AuNPs/GO but without adding GO during the preparation process) for comparative characterization. According to the preparation process, an aqueous GO solution with a similar proportion to that in the composite solution (19.23 µg/mL) and a physically mixed solution of GO and pure AuNPs (1 mg/mL GO solution to pure AuNP solution in a volume ratio of 0.0196:1) were prepared. For UV–Vis absorbance detection (200–800 nm), 2 mL of each solution was analyzed, followed by EIS at 100–1MHz using a three-electrode system (amplitude: 5 mV).

### Study on insulin response characteristics of AuNPs/GO

Given the comparable structure and functionality of bovine insulin and human insulin, and considering the relative safety and accessibility of bovine insulin, this study utilized bovine insulin to prepare pure insulin solutions for investigating the insulin response characteristics of AuNPs/GO [34, 35]. Insulin standard solutions using bovine insulin were prepared across concentrations ranging from 1.5 to 30 µM, with deionized water as the solvent. Each standard solution and 0.1 mL of deionized water were separately mixed with 1 mL of AuNPs/GO composite (prepared with 1.8 mL of reducing agent) and oscillated for 1 min. UV–Vis spectroscopy was conducted in the 200–800 nm range, with measurements repeated at least five times for each solution.

To investigate the response characteristics of various AuNPs/GO nanocomposites to bovine insulin solutions, different composites were prepared using varying volumes of reducing agents (1.5, 1.6, 1.7, 1.8, 1.9, 2.0, 2.3, and 2.5 mL). Each composite was then tested with bovine insulin solutions of differing concentrations (ranging from 1.5 to 30 µM) prepared in deionized water. The control group involved detecting deionized water before each insulin test.

Matlab 2021b was employed to calculate the visible absorption peaks in all spectra and visible absorption peak-concentration curves were generated by Excel 2021. Deionized water was used as a control group before each insulin test, and the shifts in visible absorption peaks resulting from insulin tests were determined by subtracting the visible absorption peak wavelength of the control group from the detected visible absorption peak wavelength.

### Study on commercial insulin response characteristics of AuNPs/GO

Using deionized water as the solvent, solutions of five commercial insulin products (NovoRapid®, Novolin® R, Novolin® N, Levemir®, and Novolin® 30R) were prepared at concentrations ranging from 5 to 25 µM. Then, 0.1 mL of deionized water was mixed and each of these commercial insulin solutions with 1 mL of AuNPs/GO was prepared using 2.0 mL of reducing agent, separately. After shaking for 1 min using a Lab Dancer, UV–Vis absorbance spectroscopic analysis of the mixed solutions was performed in the range of 200–800 nm. Subsequently, the corresponding concentration–displacement curves for the visible peaks were plotted. The calculation method was the same as **Section: Study on insulin response characteristics of AuNPs/GO**. Each solution was measured five times.

Using 12 μM Levemir® solutions as "unknown" test solutions, deionized water and this unknown solution were estimated by optical detection and the obtained Levemir® response curve obtained earlier. This process simulated commercial insulin detection using AuNPs/GO in a real-world scenario. 0.1 mL of water and the test solution were mixed separately with 1 mL of AuNPs/GO (prepared by adding a reducing agent of 2.0 mL) and shaken for 1 min. UV–Vis spectroscopy was performed in the range of 200–800 nm at least five times for each. Based on the visible peak shift obtained, concentrations of the test solutions were calculated and compared with true concentrations to obtain the detection error of AuNPs/GO for insulin of unknown concentration.

## Results

### Preparation and characterization of AuNPs/GO

Under the combined influence of heat and sodium citrate, tetrachloroauric acid undergoes reduction to form AuNPs. Our investigative findings revealed that graphene oxide (GO) introduction marginally expedites the reduction process when compared to the synthesis of pristine AuNPs. During the preparation of AuNPs/GO composite and pure AuNPs solution, the characteristic color transition of the AuNPs/GO mixture was observed to occur 5–10 minutes earlier than that of the pure AuNPs solution. This acceleration could be attributed to the flake-like structure of GO, which provides nucleation sites for the AuNPs, thereby reducing their nucleation time. Analysis via transmission electron microscopy (TEM) revealed that the AuNPs exhibited localized aggregation on the GO sheets with an irregular distribution, although the size of individual gold nanoparticle of the nanocomposites is relatively consistent (Fig 1A). Compared to the TEM image of pure AuNPs (S1 Fig), it can be observed that in the presence of GO, AuNPs tended to distribute on the GO nanosheets rather than aggregate. Given the uneven distribution of oxygen-containing functional groups on GO sheets, it was speculated that the AuNPs may nucleate and grow at sites of oxygen-containing functional groups on the 2D sheets in the presence of GO, leading to the faster formation of the product compared to pure AuNPs, ultimately yielding stable AuNPs/GO composites.

The AuNPs/GO composite exhibits distinct properties compared to pure AuNPs. In Fig 1B, the photos show solutions of pure AuNPs, AuNPs/GO, a physical mixture of GO and pure AuNPs labeled as "GO+AuNPs", and diluted GO from left to right. It can be seen that the color of the composite solution is darker than that of pure AuNPs and also displays a slight color difference compared to the physically mixed solution, although this difference is not easily discernible to the naked eye. Therefore, UV–Vis absorption spectroscopy was employed to differentiate between these solutions. As shown in Fig 1C, AuNPs/GO exhibit absorption peaks at 247.95 and 523.33 nm. The diluted GO solution displays an absorption peak at 242.49 nm. The pure AuNP solution presented absorption peaks at 250.49 and 520.93 nm. Furthermore, the physically mixed solution of GO and AuNPs shows absorption peaks at 248.31 and 519.9 nm. Additionally, the graph includes the sum of UV–Vis absorbance spectra of the GO solution and pure AuNP solution, serving as the theoretical UV–Vis absorbance curve of the mixture of pure GO and AuNP solution (GO+AuNPs(calculated)), with absorption peaks at 244.67 and 520.93 nm. Notably, while the UV absorption peaks of AuNPs/GO and GO+AuNPs are similar, the visible absorption peak of AuNPs/GO differs by 3 nm compared to GO+AuNPs, AuNPs, and GO+AuNPs (calculated). According to the research by Haiss et al., this difference confirms that GO/AuNPs differ from other solutions, particularly GO+AuNPs [36]. This suggests that the GO sheets and AuNPs in the AuNPs/GO solution are not merely physically mixed but rather participate in a reduction reaction during the preparation process of AuNPs/GO, consistent with the TEM results. Additionally, the size of the AuNPs can be

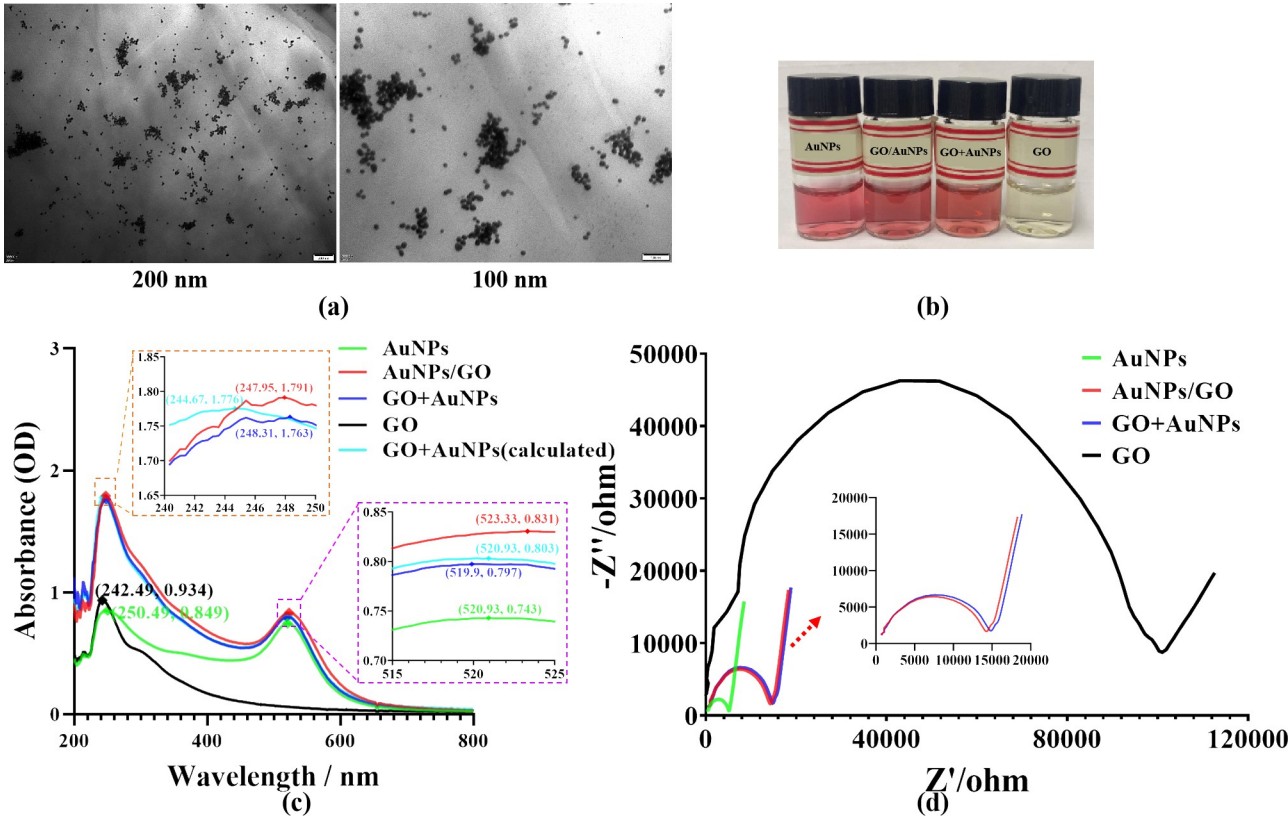

**Fig 1. Optical and electrochemical characterization of graphene oxide and gold nanoparticle composites (AuNPs/GO, prepared with 1.8 mL of reducing agent), compared with pure gold nanoparticles (AuNPs, prepared with 1.8 mL of reducing agent), diluted graphene oxide (GO), and the physical mixture of GO and pure AuNPs (marked as 'GO+AuNPs' in the figure) solutions.** (**a**) TEM images of AuNPs/GO (Scale: 200 and 100 nm). (**b**) Photos of pure AuNPs, AuNPs/GO, physical mixture of GO and pure AuNPs (GO+AuNPs), and diluted GO. (**c**) UV–Vis absorption spectroscopy characterization of all the aforementioned solutions, along with the theoretical UV–Vis absorbance curve of the mixture of pure GO and AuNPs solution (marked as 'GO+AuNPs (calculated)' in the figure). (**d**) Electrochemical impedance spectroscopy characterization of all the aforementioned solutions. The insert was the magnified image of AuNPs/GO and 'GO+AuNPs' solutions.

estimated based on the wavelength of their visible absorption peak [34]. Based on the wavelength of their visible absorption peaks, the nanoparticle size of pure AuNPs is approximately 13 nm, while the AuNP size in AuNPs/GO is approximately 18.5 nm, indicating that the presence of GO may influence the nucleation and growth of AuNPs.

Fig 1D presents a comparison of electrochemical impedance spectra of the aforementioned solutions. It is noticeable that the Nyquist curve radius of GO being nonconductive is the largest, indicating the highest impedance. By contrast, the Nyquist curve radius of pure AuNPs is the smallest, indicating the lowest impedance. The Nyquist curve of AuNPs/GO has a smaller radius compared to the physical mixture of GO and pure AuNP solution. This suggests that the sizes of AuNPs in these two solutions are similar. Hence, it is speculated that this difference may result from the nucleation of AuNPs on the oxygen functional groups on GO sheets, resulting in a decrease in the oxygen functional groups and a decrease in the impedance of AuNPs/GO composites. The primary reason for the poor conductivity of GO is the presence of oxygen functional groups. The nucleation of AuNPs on the oxygen functional groups on GO resulted in a decrease in these functional groups, which, combined with the increase in AuNPs on GO sheets, leads to a decrease in the impedance of AuNPs/GO.

These characterizations collectively demonstrate that stable composites of AuNPs/GO with strong bonds can be obtained through the preparation method described in this paper. Gold

nanoparticles in AuNPs/GO nucleates on the GO sheets, which not only reduces aggregation in solution but also further increases the specific surface area and inter-sheet spacing of GO. The synergistic interaction between AuNPs and GO manifests dual beneficial effects: Firstly, the exceptional electrical conductivity of AuNPs effectively enhances the electrical performance of graphene oxide while simultaneously suppressing nanoparticle aggregation. Secondly, the intrinsic optical properties of AuNPs and GO converge to generate composite characteristics that significantly diverge from the properties of individual nanomaterials or their physical admixture.

## Preliminary exploration of insulin response performance of AuNPs/GO

After the preparation of AuNPs/GO, this study delved into the insulin response capabilities of the composites. Previous studies on the response characteristics of AuNPs to proteins have indicated that when AuNPs interact with proteins, their visible absorption peak change (such as peak shift or absorbance change) [35, 36]. Pure insulin aqueous solutions with varying concentrations were mixed with AuNPs/GO (prepared with a reducing agent of 1.8 mL) and oscillated to obtain stable solutions. The UV–Vis absorption spectra of these mixtures are shown in Fig 2. It is evident that upon the addition of insulin, the visible absorption peak of AuNPs/GO

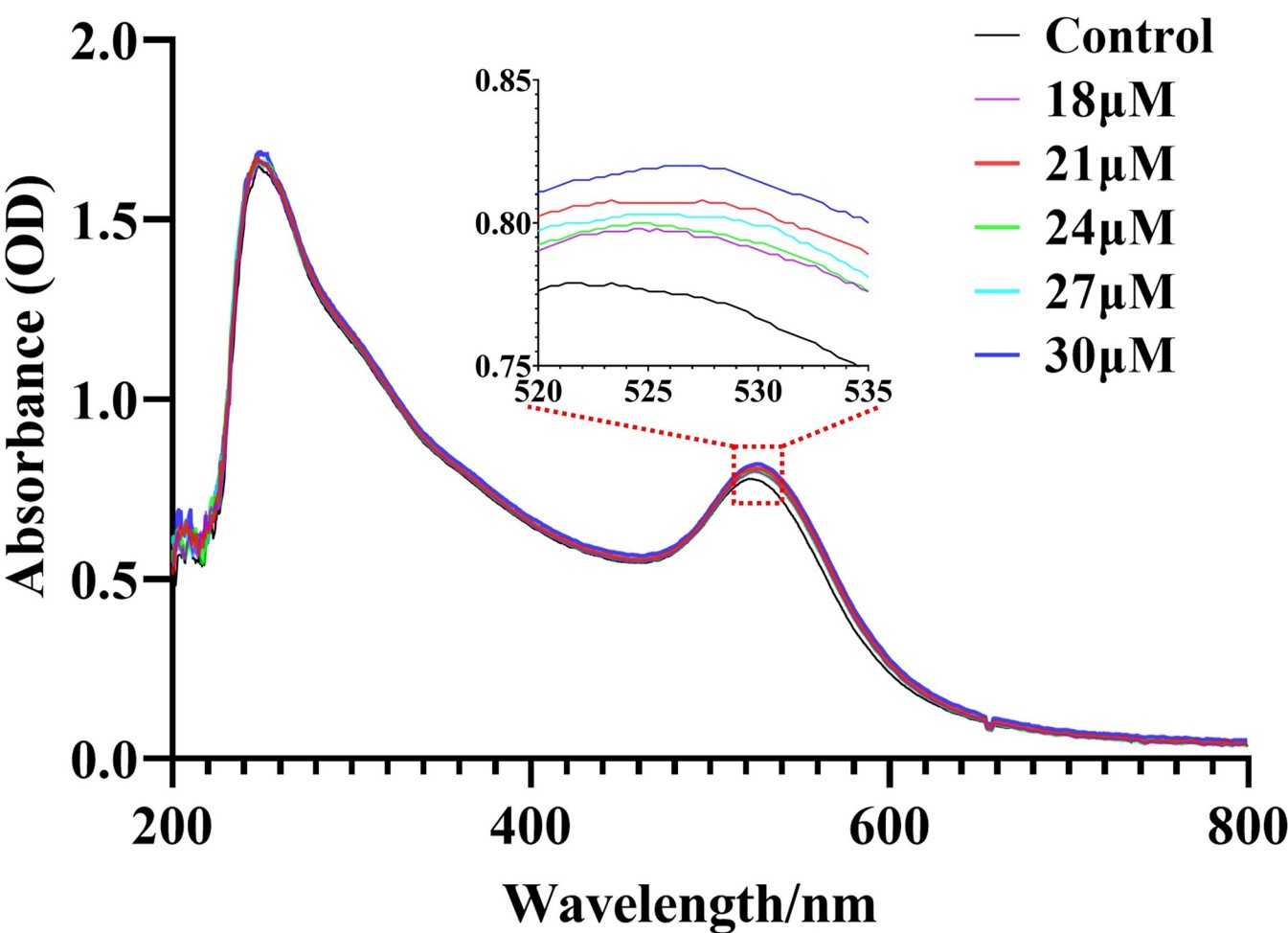

**Fig 2. UV-Vis absorption response curves of AuNPs/GO (prepared with reducing agent 1.8 mL) to pure insulin solutions at different concentrations (18–30 μM).**

shifted toward the infrared direction (rightward), with this shift becoming more pronounced with increasing insulin concentration. By contrast, the UV absorption peak of the composites primarily displayed irregular fluctuations in absorbance with minimal peak shift. This indicates that AuNPs/GO are capable of reacting with insulin, and this reaction primarily occurs at the position of AuNPs.

### Study on insulin response characteristics of different AuNPs/GO

The above analysis indicated that the insulin response characteristics of AuNPs/GO were closely related to AuNPs on the composites. Therefore, it was inferred that altering the size of AuNPs in AuNPs/GO would not only impact the optical properties of the composites but also influence their insulin-responsive results. According to previous literature reports, modifying the ratio of tetrachloroauric acid to sodium citrate in the process of preparing AuNPs could change the resulting particle size [37, 38]. Therefore, in this study, the size of AuNPs in AuNPs/GO was varied by adjusting the amount of the reducing agent (1.5–2.5 mL) during preparation, while keeping other conditions constant. And the UV-Vis spectral response characteristics of these different AuNPs/GO were investigated.

Fig 3A showed visible absorption peak wavelength–concentration-response curves of these AuNPs/GO to insulin solutions at different concentrations. It could be seen that although AuNPs/GO could respond to insulin, only a subset of AuNPs/GO exhibited concentration-dependent responses. Besides, by comparing the response curves in Fig 3A, the most suitable AuNPs/GO (prepared with reducing agent 2.0 mL) for insulin detection was identified, which exhibited the most stable and sensitive responses to insulin with a detection limit of 15.33 μM (3SD/slope). This suggested that the primary factor influencing the insulin response capability of AuNPs/GO might be the AuNPs in the composites, consistent with the above inference.

Subsequently, two different spectral response analysis methods for insulin response curve of this AuNPs/GO were further explored: The visible absorption peak-concentration response

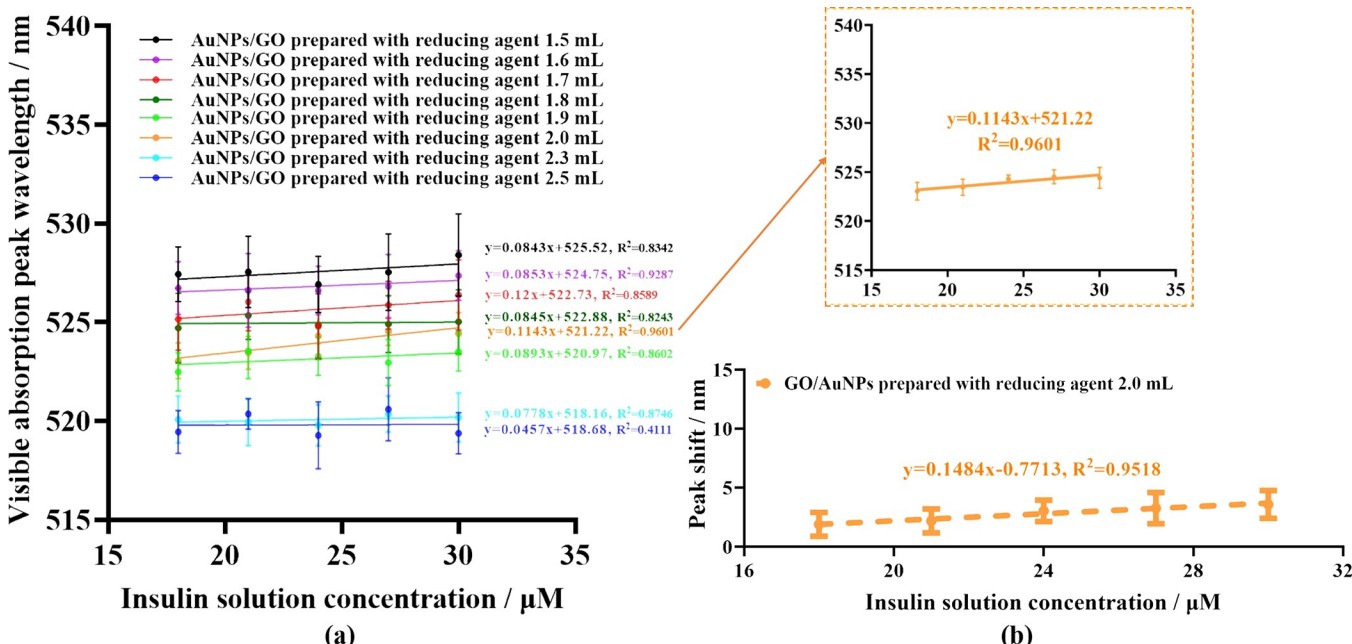

**Fig 3. Concentration gradient response curves of AuNPs/GO toward different concentrations of insulin solutions.** (**a**) Visible absorption peak wavelength–concentration-response curves to insulin by AuNPs/GO prepared with different doses of the reducing agent. (**b**) Visible absorption peak shift–concentration-response curve of AuNPs/GO prepared with 2.0 mL of reducing agent.

curve (an enlarged view of Fig 3A), and the visible absorption peak shift-concentration response curve (Fig 3B). The results showed that the slope of the absorption peak shift-concentration response curve was higher, indicating greater sensitivity with this method. And using the visible absorption peak shift as the response parameter, the calculated detection limit (3.76 μM, 3SD/slope) was lower than that achieved using the visible absorption peak wavelength. Since the R-squared values of both methods are similar, visible absorption peak shift was adopted as the response parameter in subsequent studies.

## Study on commercial insulin response capability of AuNPs/GO

This study further investigated the response characteristics of AuNPs/GO to commercial insulin, considering the differences between commercial insulin and pure insulin. Commercial insulin commonly used in clinical diagnosis and treatment falls into five categories based on onset time and duration of action: rapid-acting insulin, short-acting insulin, intermediate-acting insulin, long-acting insulin, and premixed insulin [39, 40]. These insulins not only have different main ingredients compared to pure insulin but also contain excipients such as glycerol, phenol, and disodium hydrogen phosphate. Given the varying components of different commercial insulins, it is anticipated that AuNPs/GO will exhibit different responses to these insulins. One representative drug from each of the five aforementioned types of commercial insulin was selected (NovoRapid®, Novolin® R, Novolin® N, Levemir®, and Novolin® 30R), and optical detection of these drugs was conducted using the composites.

Based on the results from **Section: Study on insulin response characteristics of different AuNPs/GO**, the responsiveness of AuNPs/GO, prepared with 2.0 mL of the reducing agent, to representative commercial insulins ranging from 5 to 25 μM was evaluated. The results are illustrated in Fig 4 and summarized in Fig 5. Fig 4 showcases that AuNPs/GO are capable of responding to all five categories of commercial insulins, but the response characteristics varied. Regarding response stability, the composites exhibited the most stable responses to Levemir®, followed by NovoRapid® and Novolin® R. In terms of response sensitivity, the composites

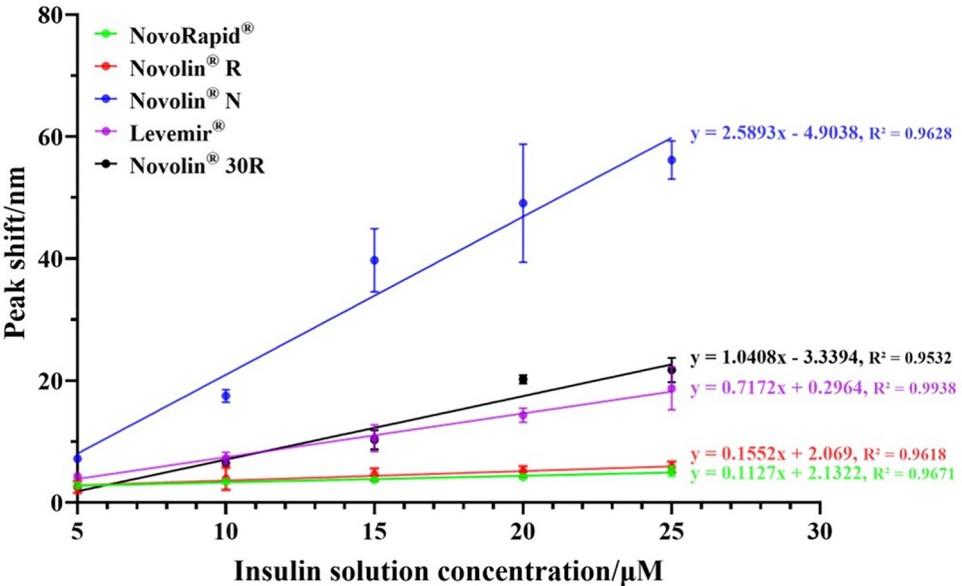

**Fig 4. Visible absorption peak shift–concentration-response curves of AuNPs/GO prepared with 2 mL reducing agent for different concentrations and types of commercial insulin.**

| Type of commercial insulin | Name of representative commercial insulin | Composition of representative commercial insulin | Characteristic features of representative commercial insulin | Limit of detection (LOD) |
|---|---|---|---|---|
| **Rapid-acting insulin** | NovoRapid® | Doorbell insulin, Guaiacol, Phenol, Glycerol, Zinc chloride, Disodium hydrogen phosphate, Sodium chloride, Hydrochloric acid/Sodium hydroxide, Sterile water for injection | Take effect 10-20 minutes after subcutaneous injection, reach peak activity in 1-3 hours, have a duration of action of 3-5 hours; Have a fast onset and short duration of action, which can lead to hypoglycemic reactions. | 4.9572 µM |
| **Short-acting insulin** | Novolin® R | Human insulin, Zinc chloride, Glycerol, Guaiacol, Sodium hydroxide, Hydrochloric acid, Sterile water for injection | Take effect within 0.5 hours after administration, reach peak activity in 1.5-3.5 hours, have a duration of action of about 7-8 hours; Can be administered intravenously. | 3.5997 µM |
| **Intermediate-acting insulin** | Novolin® N | Human insulin, Protamine sulfate , Zinc chloride, Glycerol, Disodium hydrogen phosphate, Methyl salicylate, Phenol, Hydrochloric acid, Sodium hydroxide, Sterile water for injection | Take effect within 1.5 hours after administration, reach peak activity in 4-12 hours, have a duration of action of approximately 24 hours; Generally administered via subcutaneous injection. | 0.2158 µM (detection by AuNPs/GO prepared using 2 mL reducing agent); 2.9641µM (detection by AuNPs/GO prepared using 2.3 mL reducing agent) |
| **Long-acting insulin** | Levemir® | Detemir insulin, Methyl salicylate, Phenol, Glycerol, Zinc acetate, Disodium hydrogen phosphate, Sodium chloride, Hydrochloric acid, Sodium hydroxide, Sterile water for injection | Take effect within 1.5 to 2 hours, reach peak activity in 6 to 8 hours, have a duration of action of approximately 18 to 24 hours, or even longer; Show a prolonged duration of action, cannot be administered intravenously. | 0.7790 µM |
| **Pre-mixed insulin** | Novolin® 30R | Isophane Insulin (70%), Soluble neutral insulin (30%), Protamine sulfate, Zinc chloride, Glycerol, Disodium hydrogen phosphate, Methyl salicylate, Phenol, Hydrochloric acid, Sodium hydroxide, Sterile water for injection | Short-acting or rapid-acting insulin is mixed with intermediate-acting insulin in a certain proportion; Take effect within 30 minutes with the short-acting insulin reaching peak activity in 1.5 to 2.5 hours, have a duration of action of approximately 16 to 24 hours; Can rapidly lower blood sugar levels, have a long duration of action. | 0.5368 µM |

**Fig 5. Composition, characteristics, and limits of detection (LOD, 3SD/slope) by AuNPs/GO (prepared with 2 mL of reducing agent) for representative commercial insulin samples in this study.**

demonstrated the highest responses to Novolin® N, which were even observable with the naked eye (Fig 6A), followed by Novolin® 30R. Concerning the limit of detection (LOD, calculated as 3SD/slope), AuNPs/GO exhibited the lowest LOD for Novolin® N, which was 0.2158 µM, followed by Novolin® 30R (Fig 5). It is evident that the composites displayed stronger responses to Novolin® N and Novolin® 30R. Considering their components (Fig 5), these two insulins contain protamine sulfate, which is rich in glutamic acid and can also interact with the composites, thereby enhancing the optical response signals.

Conversely, as shown in Fig 4, the responses of AuNPs/GO to Novolin® N, particularly at high concentrations, were highly unstable. Therefore, this study also explored the response outcomes of AuNPs/GO prepared with 2.3 mL of the reducing agent to concentrations ranging from 5–25 µM Novolin® N. Fig 6B presents a comparison of response curves between the two types of AuNPs/GO (prepared with 2 and 2.3 mL of reducing agent, respectively) to Novolin® N. It is evident that with an increase in the amount of reducing agent during preparation, the response sensitivity of AuNPs/GO to Novolin® N notably decreased, and the limit of detection (LOD) increased from 0.2158 to 2.9641 µM. However, the response stability of the composites significantly improved.

To explore the practical application potential of AuNPs/GO, a Levemir® solution at 12 µM was also detected as an 'unknown concentration' insulin solution. By mixing it with AuNPs/GO prepared with 2.0 mL of reducing agent, the obtained optical response results were fitted into the response curve of Levemir® shown in Fig 4. An estimated concentration of 12.19688 µM was derived, with an error of less than 2% compared to the actual concentration

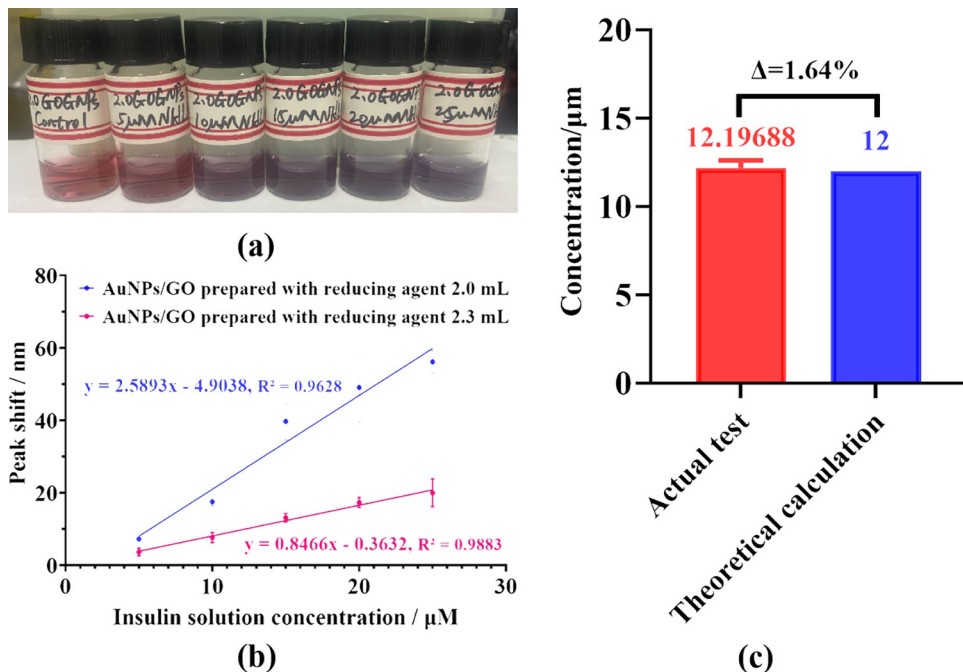

**Fig 6. Optical response results of AuNPs/GO to commercial insulin solutions.** (**a**) Photograph of AuNPs/GO mixed with different concentrations of Novolin[®] N. (**b**) Visible absorption peak shift–concentration-response curves for different concentrations of Novolin[®] N using AuNPs/GO prepared with 2 and 2.3 mL of reducing agents, respectively. (**c**) Concentration testing result of the Levemir[®] solution at 'unknown concentration' using AuNPs/GO compared with its actual concentration.

(Fig 6C), demonstrating the potential of the composites in insulin sensing. In summary, AuNPs/GO can exhibit responsiveness to various commercial insulins, and changing the preparation condition of AuNPs/GO can result in different responses to these insulins.

## Discussion

The results exhibit that AuNPs/GO can really response to pure and commercial insulins. Based on our previous research, AuNPs/GO exhibited optical responses only to cysteine through sulfur-hydrogen bonds, and acidic amino acids by the interactions betweenα-amino acids on them and AuNPs, among essential amino acids. Cysteine forms a gold–sulfur bond with the composites through its thiol group, while the acidic amino acids most likely react with AuNPs/GO through α-carboxyl groups [33]. Insulin contains six cysteines and four glutamic acids among its 51 amino acids, as well as three asparagines and three glutamines. The insulin responsiveness of AuNPs/GO may be attributed to interactions with specific amino acid residues containing α-carboxyl groups, namely asparagine and glutamine, which exhibit similar chemical properties to aspartic acid and glutamic acid. Furthermore, while insulin contains three pairs of cysteine residues forming intramolecular disulfide bonds, these native bonds may be cleaved by residual reducing agents present in the AuNPs/GO composite solution. This reduction potentially enables direct binding between the exposed cysteine residues and the gold nanoparticles through stable gold-sulfur interactions [41]. Thus, the molecular recognition between AuNPs/GO and insulin likely occurs through dual mechanisms: interactions with carboxyl-containing residues and coordination via reduced cysteine residues.

The structural composition of AuNPs/GO exhibits significant variation depending on preparation conditions. These variations manifest not only in the diameter of AuNPs deposited on

GO sheets but also in their spatial distribution and aggregation patterns. Such structural characteristics fundamentally influence the optical properties of AuNPs/GO composites and their subsequent interactions with biomolecules, particularly large protein molecules such as insulin. While AuNPs of smaller diameter theoretically offer enhanced detection sensitivity, their diminutive size may lead to excessive packing density on GO surfaces, potentially restricting the available surface area for insulin interaction and consequently attenuating the response magnitude. Conversely, larger AuNP diameters result in diminished sensitivity, ultimately compromising insulin detection capabilities. These theoretical considerations are corroborated by both direct and indirect experimental evidence presented in this investigation. Thus, prior to conducting application-oriented studies, it is imperative to establish optimal preparation parameters for AuNPs/GO to ensure maximum detection efficacy. The varying responses of AuNPs/GO to different commercial insulins suggested that, although excipients represent a small proportion, their small molecular size allowed them to interact with AuNPs/GO through interactions between thiol or α-carboxyl groups and the composites, significantly affecting the spectral responses. This ultimately leaded to differences in the responses of AuNPs/GO to various commercial insulins. On one hand, this provided an opportunity to distinguish between different types of commercial insulin; on the other hand, it highlighted that the specificity of AuNPs/GO needed to be improved. Moreover, a high concentration (2 mg/mL) of insulin was also mixed with AuNPs/GO, and the spectrum was detected accordingly. The spectrum demonstrated a shift in the absorbance peak instead of the appearance of another peak at 680 nm, as observed in amino acid detection by AuNPs/GO. This suggests that the small size of amino acid molecules enables them to aggregate on the composite's surface, disrupting their charge balance and resulting in multiple absorption peaks in the visible light region. By contrast, insulin molecules are larger and may bind to multiple composite molecules around them. The presence of insulin molecules maintains the gold nanoparticles on different composite sheets at a certain distance, preventing the aggregation of AuNPs/GO. Therefore, the composites retain the local surface plasmon resonance peak of AuNPs when reacting with insulin.

Furthermore, based on the responsive results of different AuNPs/GO to commercial insulin, it can be speculated that AuNPs on GO nanosheets gained a reduced binding capacity with large molecular proteins and better stability compared to that of pure nanoparticles, because of the addition of GO sheets.

## Conclusion

This study explored the response characteristics of different AuNPs/GO composites to biomacromolecules, particularly insulin, by varying the preparation conditions. The research not only investigated the responsiveness of the composites to pure insulin but also examined its response characteristics to the representative products of the five commercial insulin categories. The results demonstrated that AuNPs/GO could respond to insulin, and this response was likely caused by the interactions between the α-amino acids on the protein molecules and the AuNPs in the composites. The response capability of AuNPs/GO toward insulin varies under different preparation conditions, suggesting differences in the properties of the composites owing to variations in the size of AuNPs in the composites. The composites exhibited responses to all five categories of commercial insulin with distinct characteristics and demonstrated the ability to accurately determine unknown concentrations of commercial insulin, confirming their potential applications in quality control during insulin production and in differentiating between insulin types. In summary, this study provided an experimental foundation for advancing applications of AuNPs/GO composites in biomolecule sensing, and verified that such nanocomposites could show different responsiveness by altering preparation

conditions. Moreover, it demonstrated the application potential of AuNPs/GO in areas such as biomacromolecule and drug testing and production.

## Supporting information

**S1 Fig. TEM images of pure AuNPs (Scale: 100 nm), which was taken by JEM-1230 (JEOL, 120kV).**
(TIF)

**S1 Dataset. All data associated with the figures in this manuscript (except for the TEM images).**
(XLSX)

**S1 Code. MATLAB code for batch converting txt files into mat files, and extracting the absorption peak wavelengths and peak values from response curves of different insulin samples.**
(DOCX)

## Author Contributions

**Conceptualization:** Qian Zhang, Jingyi Feng.

**Data curation:** Qian Zhang, Yanjun Pan, Jin Pan.

**Formal analysis:** Qian Zhang, Yanjun Pan, Jingyi Feng.

**Funding acquisition:** Qian Zhang.

**Investigation:** Jin Pan, Jingyi Feng.

**Methodology:** Qian Zhang.

**Project administration:** Jingyi Feng.

**Supervision:** Jingyi Feng.

**Writing – original draft:** Qian Zhang.

**Writing – review & editing:** Zhichen Wang, Ruyi Lu, Jing Sun.

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
