## [Decision Letter · Decision Letter 0]

22 Oct 2024

PONE-D-24-28170Differential insulin response characteristics of graphene oxide–gold nanoparticle composites under varied synthesis conditionsPLOS ONE

Dear Dr. Feng,

Thank you for submitting your manuscript to PLOS ONE. After careful consideration, we feel that it has merit but does not fully meet PLOS ONE’s publication criteria as it currently stands. Therefore, we invite you to submit a revised version of the manuscript that addresses the points raised during the review process.

We look forward to receiving your revised manuscript.

Kind regards,

Munna Lal Yadav, Ph.D

Academic Editor

PLOS ONE

“This research was funded by Zhejiang Provincial Natural Science Foundation of China, Grant No. LQ20C100001, and Zhejiang Provincial Department of Education General Research Project of China, Grant No. Y201941826.”

6. Please include a new copy of Table 1 in your manuscript; the current table is difficult to read. Please follow the link for more information: https://blogs.plos.org/plos/2019/06/looking-good-tips-for-creating-your-plos-figures-graphics/

7. Please include your tables as part of your main manuscript and remove the individual files. Please note that supplementary tables (should remain/ be uploaded) as separate "supporting information" files

Reviewers' comments:

Reviewer's Responses to Questions

**Comments to the Author**

1. Is the manuscript technically sound, and do the data support the conclusions?

Reviewer #1: Yes

Reviewer #2: Yes

2. Has the statistical analysis been performed appropriately and rigorously? 

Reviewer #1: Yes

Reviewer #2: Yes

3. Have the authors made all data underlying the findings in their manuscript fully available?

Reviewer #1: Yes

Reviewer #2: Yes

4. Is the manuscript presented in an intelligible fashion and written in standard English?

Reviewer #1: Yes

Reviewer #2: Yes

5. Review Comments to the Author

Reviewer #1: The authors has worked on Pure and commercial Insulin detection by the help of AuNPs/GO and also examined its response characteristics to the representative products of the five commercial Insulin categories and demonstrated the ability to accurately determine unknown concentrations of commercial insulin, confirming their potential applications in drug sensing. This research group has wide experience on development of biological activities.

1) Page number – 10 Line number - 25

It is mentioned GO/GNPs in full article. It should be AuNPs/GO in place of GO/GNPs.

2) Page number – 16 Line number - 145

The full form of instruments are mentioned with the abbreviation but in this case of EIS it is not mentioned.

3) Chloroauric acid is costly so how should the author justify the cost-effectiveness of the synthesized product?

Reviewer #2: In this study, the aim is to explore the response characteristics of graphene oxide (GO) and gold nanoparticle (GNP) composites to insulin and insulin commercial formulations, under different preparation conditions. The manuscript fulfill all aspects mention in the questions above. However, there are some minor observations to be consider to improve the document.

The following observations and suggestions are proposed:

Introduction:

• Lines 59–61: It would be beneficial to delve deeper into the resulting composition of the GNPs and GO combination in order to highlight the specific binding sites that may serve as interaction points. It would be valuable to include original research articles that emphasize the evaluation of the sensitivity of these components.

• Line 82–83: While it is true that insulin is a highly studied hormone in the clinical setting for assessing patients' metabolic states, the project focuses on the characterization and evaluation of nanoparticles with different types of insulin. In clinical practice, a detailed analysis of different insulin types is not commonly performed. However, I would like to emphasize that in the future, distinguishing between different types of insulin in the body could become important (e.g., for monitoring dosing), but as for today, this approach may be more applicable in quality control during insulin production and in differentiating between insulin types.

Results:

• Lines 206–207: To what extent does the reduction process was accelerated? Could the quantity of nanoparticles formed be quantified?

• Figure 1a: It may be helpful to show a transmission electron microscopy (TEM) image of the gold nanoparticles without GO as a control.

• Lines 263–265: In addition to the interaction between functional groups and the gold particles, it would be beneficial to emphasize the observed distribution and integration in the electron microscopy images, highlighting how these improve the electrical connectivity compared to other mixtures.

• Lines 283–289 (Figure 2): It would be useful to provide more detail on the changes observed in the absorption wavelengths at different insulin concentrations. There does not seem to be a concentration-dependent phenomenon in

response to insulin.

Discussion:

• Lines 398–400: The addition of the reducing agent to the mixture with insulin causes the native disulfide bonds in the insulin structure to break, thus, the cysteine residues could be involved in the reaction, as well as the other amino acids discussed by the authors.

• A more in-depth discussion of the characterization of the different preparation conditions and their impact on the response to insulin would be beneficial.

• It may also be suggested to evaluate the potential influence of excipients in insulin formulations on the response of the GNPs in future studies.

6. PLOS authors have the option to publish the peer review history of their article (what does this mean?). If published, this will include your full peer review and any attached files.

Reviewer #1: **Yes: **Dr. Dilip Vasava

Reviewer #2: No

---

## [Author Response · Author response to Decision Letter 0]

6 Dec 2024

Reviewer #1: 

The authors has worked on Pure and commercial Insulin detection by the help of AuNPs/GO and also examined its response characteristics to the representative products of the five commercial Insulin categories and demonstrated the ability to accurately determine unknown concentrations of commercial insulin, confirming their potential applications in drug sensing. This research group has wide experience on development of biological activities.

1) Page number – 10 Line number - 25

It is mentioned GO/GNPs in full article. It should be AuNPs/GO in place of GO/GNPs.

Agreed.

Thank you very much for your comments. We have conducted a thorough review of the manuscript, replacing all instances of 'GNP' with 'AuNP' and all occurrences of 'GO/GNPs' with 'AuNPs/GO'. These revisions have been highlighted throughout the text.

2) Page number – 16 Line number - 145

The full form of instruments are mentioned with the abbreviation but in this case of EIS it is not mentioned.

Agreed.

Thank you for your careful comments. We first introduced EIS in Section '2.1 Materials and instruments', where the full term is provided as “Electrochemical impedance spectroscopy (EIS)”. Please refer to Line 144 in the manuscript.

3) Chloroauric acid is costly so how should the author justify the cost-effectiveness of the synthesized product?

Agreed.

Thank you for your thoughtful comments. In our study, gold nanoparticles (AuNPs) were employed at trace levels for single-instance detection purposes. The preparation of 150 mL AuNPs/GO mixture requires approximately 1.24 g of chloroauric acid, with an associated cost of approximately 2 US dollars. Given that each detection assay requires 2 mL of the AuNPs/GO mixture, and accounting for necessary control groups, a single batch (150 mL) can accommodate up to 75 independent spectroscopic measurements. This translates to a chloroauric acid cost of less than 0.03 US dollars per individual test. When implemented in conjunction with smartphone-based portable spectroscopic detection [1], we believe this methodology represents a cost-effective approach for practical applications.

Reviewer #2: 

In this study, the aim is to explore the response characteristics of graphene oxide (GO) and gold nanoparticle (GNP) composites to insulin and insulin commercial formulations, under different preparation conditions. The manuscript fulfill all aspects mention in the questions above. However, there are some minor observations to be consider to improve the document.

The following observations and suggestions are proposed:

Introduction:

• Lines 59–61: It would be beneficial to delve deeper into the resulting composition of the GNPs and GO combination in order to highlight the specific binding sites that may serve as interaction points. It would be valuable to include original research articles that emphasize the evaluation of the sensitivity of these components.

Agreed.

Thank you for your thoughtful comments. We have added three original research articles to emphasize the evaluation of the sensitivity of GNPs and GO separately and in combination (Line 61-77). Zhu et al. fabricated a periodically tapered structure-based gold nanoparticles and graphene oxide - immobilized optical fiber sensor to detect ascorbic acid, and found that use of AuNPs and GO could enhance the sensing performance of the sensor.[2] Mohsin et al. developed an innovative sensing platform employing gold nanoparticles and reduced graphene oxide (rGO) for ultrasensitive Hepatitis B virus detection. Their investigation revealed that the strategic incorporation of AuNPs with rGO provides multifaceted advantages, including preventing nanosheet structural aggregation, substantially increasing the effective surface area, significantly lowering detection limits, and markedly improving biocompatibility.[3] Yang et al. conducted a sophisticated localized surface plasmon resonance (LSPR)-based tapered fiber sensor utilizing gold nanoparticles and graphene oxide. Their research elucidated that LSPR intensity and wavelength are critically modulated by the morphological and structural characteristics of gold nanoparticles, specifically the nanoparticle size, morphological configuration, and surrounding medium's refractive index.[4, 5] Consequently, the conjugation of AuNPs with GO induces substantial modifications in sensitivity and LSPR properties, presenting promising opportunities for advanced biochemical sensing and diagnostic applications.

• Line 82–83: While it is true that insulin is a highly studied hormone in the clinical setting for assessing patients' metabolic states, the project focuses on the characterization and evaluation of nanoparticles with different types of insulin. In clinical practice, a detailed analysis of different insulin types is not commonly performed. However, I would like to emphasize that in the future, distinguishing between different types of insulin in the body could become important (e.g., for monitoring dosing), but as for today, this approach may be more applicable in quality control during insulin production and in differentiating between insulin types.

Agreed.

Thank you for your valuable comments. According to your comments, we have advised relevant expression to “this study demonstrated the application potential of AuNPs/GO in areas such as drug testing and production, (Line 40-41) ... As a basic protein, insulin is important and highly studied. (Line 97) ... The composites exhibited responses to all five categories of commercial insulin with distinct characteristics and demonstrated the ability to accurately determine unknown concentrations of commercial insulin, confirming their potential applications in quality control during insulin production and in differentiating between insulin types. (Line 493-497) ... Moreover, it demonstrated the application potential of AuNPs/GO in areas such as biomacromolecule and drug testing and production. (Line 500-502)” 

Results:

• Lines 206–207: To what extent does the reduction process was accelerated? Could the quantity of nanoparticles formed be quantified?

Agreed.

We appreciate your detailed observations regarding the AuNPs/GO synthesis process. As noted in our manuscript (Lines 222-228), we observed that the AuNPs/GO mixture achieved its final coloration 5-10 minutes earlier than the pure AuNPs solution. This acceleration, though modest, can be attributed to GO sheets providing nucleation sites for AuNPs formation. This observation serves as indirect evidence supporting our proposed mechanism of AuNPs nucleation and growth on GO sheets.

Regarding the concentration analysis, based on our preparation method detailed in Section 2.2, we calculated the mass concentration of Au to be 0.039 mg/mL (1.99×10-4 mol/L). We also explored quantifying AuNPs concentration in the AuNPs/GO complex using UV-Vis spectroscopy, following the methodology described by Haiss et al.[6] This analysis yielded an approximate concentration of 2.48×1010 particles/mL. However, we should note that while this quantification method was developed for pure AuNPs in aqueous solutions, our system involves AuNPs conjugated to GO sheets. Therefore, we have opted to provide this value solely for reference purposes and have not included it in the main manuscript.

• Figure 1a: It may be helpful to show a transmission electron microscopy (TEM) image of the gold nanoparticles without GO as a control.

Agreed.

Thank you for your thoughtful comments. We have supplemented the TEM images of pure AuNPs (prepared with 1.8 mL of reducing agent), as shown in Figure (a) below. Figure (b) corresponds to Figure 1a in our manuscript. It can be observed that in the presence of GO, AuNPs tended to distribute on the GO nanosheets rather than aggregate. (Line 232-234) Since the TEM instrument (JEM-1230, 120 kV) used for the additional imaging of pure AuNPs differs from that of AuNPs/GO, this TEM image has been placed in the supplementary materials. (Fig. S1, Supplementary Materials)

(a) (b)

• Lines 263–265: In addition to the interaction between functional groups and the gold particles, it would be beneficial to emphasize the observed distribution and integration in the electron microscopy images, highlighting how these improve the electrical connectivity compared to other mixtures.

Agreed.

Thank you for your valuable comments. In addition to the description of AuNPs' distribution and integration added in Line 232-234, we have also included the following statement in Line 295-303: “Gold nanoparticles in AuNPs/GO nucleates on the GO sheets, which not only reduces aggregation in solution but also further increases the specific surface area and inter-sheet spacing of GO. The synergistic interaction between AuNPs and GO manifests dual beneficial effects: Firstly, the exceptional electrical conductivity of AuNPs effectively enhances the electrical performance of graphene oxide while simultaneously suppressing nanoparticle aggregation. Secondly, the intrinsic optical properties of AuNPs and GO converge to generate composite characteristics that significantly diverge from the properties of individual nanomaterials or their physical admixture.”

• Lines 283–289 (Figure 2): It would be useful to provide more detail on the changes observed in the absorption wavelengths at different insulin concentrations. There does not seem to be a concentration-dependent phenomenon in response to insulin.

Agreed.

Thank you very much for your comments. Our interpretation of Fig. 3 is not sufficiently complete, leading to potential ambiguity. Although AuNPs/GO could respond to insulin, it can be seen in Fig. 3a that only a subset of AuNPs/GO samples prepared under different conditions exhibited concentration-dependent responses to insulin. Besides, by comparing the response curves in Fig. 3a, the most suitable AuNPs/GO (prepared with reducing agent 2.0 mL) for insulin detection was identified. (Line 336-340) Subsequently, two different spectral response analysis methods for insulin response curve of this AuNPs/GO were further explored: The visible absorption peak-concentration response curve (an enlarged view of Fig. 3a), and the visible absorption peak shift-concentration response curve (Fig. 3b). The results showed that the slope of the absorption peak shift-concentration response curve was higher, indicating greater sensitivity with this method. And using the visible absorption peak shift as the response parameter, the calculated detection limit (3.76 μM, 3SD/slope) was lower than that achieved using the visible absorption peak wavelength. Since the R-squared values of both methods are similar, visible absorption peak shift was adopted as the response parameter in subsequent studies. (Line 351-360)

Discussion:

• Lines 398–400: The addition of the reducing agent to the mixture with insulin causes the native disulfide bonds in the insulin structure to break, thus, the cysteine residues could be involved in the reaction, as well as the other amino acids discussed by the authors.

Agreed.

Thank you for your professional comments. We have incorporated your comments into the discussion section: “The insulin responsiveness of AuNPs/GO may be attributed to interactions with specific amino acid residues containing α-carboxyl groups, namely asparagine and glutamine, which exhibit similar chemical properties to aspartic acid and glutamic acid. Furthermore, while insulin contains three pairs of cysteine residues forming intramolecular disulfide bonds, these native bonds may be cleaved by residual reducing agents present in the AuNPs/GO composite solution. This reduction potentially enables direct binding between the exposed cysteine residues and the gold nanoparticles through stable gold-sulfur interactions. Thus, the molecular recognition between AuNPs/GO and insulin likely occurs through dual mechanisms: interactions with carboxyl-containing residues and coordination via reduced cysteine residues.” (Line 435-445)

• A more in-depth discussion of the characterization of the different preparation conditions and their impact on the response to insulin would be beneficial.

Agreed.

Thank you very much for your comments. We have added further analysis in the discussion regarding the impact of different preparation conditions of AuNPs/GO on insulin detection: “The structural composition of AuNPs/GO exhibits significant variation depending on preparation conditions. These variations manifest not only in the diameter of AuNPs deposited on GO sheets but also in their spatial distribution and aggregation patterns. Such structural characteristics fundamentally influence the optical properties of AuNPs/GO composites and their subsequent interactions with biomolecules, particularly large protein molecules such as insulin. While AuNPs of smaller diameter theoretically offer enhanced detection sensitivity, their diminutive size may lead to excessive packing density on GO surfaces, potentially restricting the available surface area for insulin interaction and consequently attenuating the response magnitude. Conversely, larger AuNP diameters result in diminished sensitivity, ultimately compromising insulin detection capabilities. These theoretical considerations are corroborated by both direct and indirect experimental evidence presented in this investigation. Thus, prior to conducting application-oriented studies, it is imperative to establish optimal preparation parameters for AuNPs/GO to ensure maximum detection efficacy.” (Line 446-460)

• It may also be suggested to evaluate the potential influence of excipients in insulin formulations on the response of the GNPs in future studies.

Agreed.

Thank you for your thoughtful comments. We have added some discussion on the influence of excipients in commercial insulins: “The varying responses of AuNPs/GO to different commercial insulins suggested that, although excipients represent a small proportion, their small molecular size allowed them to interact with AuNPs/GO through interactions between thiol or α-carboxyl groups and the composites, significantly affecting the spectral responses. This ultimately leaded to differences in the responses of AuNPs/GO to various commercial insulins. On one hand, this provided an opportunity to distinguish between different types of commercial insulin; on the other hand, it highlighted that the specificity of AuNPs/GO needed to be improved.” (Line 460-467)

References

1. Chen W, Yao Y, Chen T, Shen W, Tang S, Lee HK. Application of smartphone-based spectroscopy to biosample analysis: A review. Biosensors and Bioelectronics. 2021;172:112788.

2. Zhu G, Agrawal N, Singh R, Kumar S, Zhang B, Saha C, et al. A novel periodically tapered structure-based gold nanoparticles and graphene oxide–Immobilized optical fiber sensor to detect ascorbic acid. Optics & Laser Technology. 2020;127:106156.

3. Mohsin DH, Mashkour MS, Fatemi F. Design of aptamer-based sensing platform using gold nanoparticles functionalized reduced graphene oxide for ultrasensitive detection of Hepatitis B virus. Chemical Papers. 2021;75(1):279-95.

4. Yang Q, Zhu G, Singh L, Wang Y, Singh R, Zhang B, et al. Highly sensitive and selective sensor probe using glucose oxidase/gold nanoparticles/graphene oxide functionalized tapered optical fiber structure for detection of glucose. Optik. 2020;208:164536.

5. Kim H-M, Jeong DH, Lee H-Y, Park J-H, Lee S-K. Improved stability of gold nanoparticles on the optical fiber and their application to refractive index sensor based on localized surface plasmon resonance. Optics & Laser Technology. 2019;114:171-8.

6. Haiss W, Thanh NT, Aveyard J, Fernig DG. Determination of size and concentration of gold nanoparticles from UV− Vis spectra. Analytical chemistry. 2007;79(11):4215-21.

7. Li Y, Gong H, Sun Y, Yan J, Cheng B, Zhang X, et al. Dissecting the role of disulfide bonds on the amyloid formation of insulin. Biochemical and biophysical research communications. 2012;423(2):373-8.

https://j

---

## [Decision Letter · Decision Letter 1]

22 Dec 2024

Differential insulin response characteristics of graphene oxide–gold nanoparticle composites under varied synthesis conditions

PONE-D-24-28170R1

Dear Dr. Feng,

We’re pleased to inform you that your manuscript has been judged scientifically suitable for publication and will be formally accepted for publication once it meets all outstanding technical requirements.

Kind regards,

Munna Lal Yadav, Ph.D

Academic Editor

PLOS ONE

Additional Editor Comments (optional):

Reviewers' comments:

Reviewer's Responses to Questions

**Comments to the Author**

1. If the authors have adequately addressed your comments raised in a previous round of review and you feel that this manuscript is now acceptable for publication, you may indicate that here to bypass the “Comments to the Author” section, enter your conflict of interest statement in the “Confidential to Editor” section, and submit your "Accept" recommendation.

Reviewer #1: All comments have been addressed

Reviewer #2: All comments have been addressed

2. Is the manuscript technically sound, and do the data support the conclusions?

Reviewer #1: Yes

Reviewer #2: Yes

3. Has the statistical analysis been performed appropriately and rigorously? 

Reviewer #1: Yes

Reviewer #2: Yes

4. Have the authors made all data underlying the findings in their manuscript fully available?

Reviewer #1: Yes

Reviewer #2: Yes

5. Is the manuscript presented in an intelligible fashion and written in standard English?

Reviewer #1: Yes

Reviewer #2: Yes

6. Review Comments to the Author

Reviewer #1: (No Response)

Reviewer #2: Thank you for consideriong and adequately addressing all the comments rised in the prevoius revivision

7. PLOS authors have the option to publish the peer review history of their article (what does this mean?). If published, this will include your full peer review and any attached files.

Reviewer #1: **Yes: **Dilip V Vasava, Department of Chemistry, Gujarat University, Ahmedabad, Gujarat, India

Reviewer #2: No

---

## [Editor Report · Acceptance letter]

26 Dec 2024

PONE-D-24-28170R1 

PLOS ONE

Dear Dr. Feng, 

I'm pleased to inform you that your manuscript has been deemed suitable for publication in PLOS ONE. Congratulations! Your manuscript is now being handed over to our production team.

Kind regards, 

on behalf of

Dr. Munna Lal Yadav 

Academic Editor

PLOS ONE